# Ferric Carboxymaltose Improves the Quality of Life of Patients with Inflammatory Bowel Disease and Iron Deficiency without Anaemia

**DOI:** 10.3390/jcm11102786

**Published:** 2022-05-15

**Authors:** Jose María Huguet, Xavier Cortés, Marta Maia Boscá-Watts, Margarita Muñoz, Nuria Maroto, Marisa Iborra, Esther Hinojosa, María Capilla, Carmina Asencio, Cirilo Amoros, Jose María Paredes

**Affiliations:** 1Gastroenterology Department, Hospital General Universitario de Valencia, 46014 Valencia, Spain; capigumar@gmail.com; 2Gastroenterology Department, Hospital de Sagunto, 46520 Valencia, Spain; xacori@gmail.com; 3Gastroenterology Department, Hospital Clínico Universitario de Valencia, University of Valencia, 46010 Valencia, Spain; maiabosca@gmail.com; 4Gastroenterology Department, Hospital General Universitario de Castellón, 12004 Castesllon de la Plana, Spain; munyoz_marvice@yahoo.es; 5Gastroenterology Department, Hospital de Manises, 46940 Valencia, Spain; nuriamaroto2002@hotmail.com (N.M.); hinova200@gmail.com (E.H.); 6Gastroenterology Department, Hospital Universitario y Politécnico de La Fe, 46026 Valencia, Spain; marisaiborra@hotmail.com; 7Gastroenterology Department, Hospital Universitario Doctor Peset, 46017 Valencia, Spain; carmina.asencio@gmail.com (C.A.); chemaparedes1969@gmail.com (J.M.P.); 8Gastroenterology Department, Hospital Arnau de Vilanova de Valencia, 46015 Valencia, Spain; camorosg@gmail.com

**Keywords:** iron deficiency, health-related quality of life, ferric carboxymaltose, inflammatory bowel disease

## Abstract

Background: Iron deficiency (ID) without anaemia is a common comorbidity associated with inflammatory bowel disease (IBD) that has a negative impact on health-related quality of life (HRQoL). Methods: This multicentre, prospective, observational study examined the response to, safety of and impact on HRQoL of a single 500 mg dose of intravenous ferric carboxymaltose (FCM) in patients with IBD and ID without anaemia. The diagnostic criteria for ID were low serum ferritin (<30 µg/L in the absence of inflammatory activity or <100 µg/L with inflammation) and transferrin saturation index (TSAT) < 16%. The effect on iron levels and HRQoL, according to the health status questionnaires SF-12v2 and EQ-5D, was evaluated 1 month after FCM infusion in an outpatient setting. Results: Of the 105 patients who received FCM, 98 patients completed the study. After 1 month, a single dose of FCM significantly increased serum ferritin, serum iron and TSAT. Importantly, patients reported fewer ID symptoms and problems on all EQ-5D dimensions. They also had higher EQ-5D visual analogue scale and SF-12v2 scores after treatment. FCM had similar clinical effects on men and women and on patients with Crohn’s disease (n = 66) and ulcerative colitis (n = 32). Conclusion: A single dose of FCM rapidly restored iron parameters and significantly improved patients’ symptoms and HRQoL at 1 month after treatment.

## 1. Introduction

Inflammatory bowel disease (IBD) refers to a group of chronic relapsing inflammatory disorders, including Crohn’s disease (CD) and ulcerative colitis (UC), with a rising worldwide incidence and prevalence [1,2]. Anaemia is the most common extra-intestinal complication in IBD [3], and approximately one in five patients with IBD is anaemic at any given time [4,5]. In over 50% of these patients, anaemia is caused by iron deficiency (ID) [6]. Intravenous administration of ferric carboxymaltose (FCM) is an effective and safe treatment for restoring haemoglobin (Hb) levels and significantly improving the health-related quality of life (HRQoL) of these patients [7,8,9].

Anaemia is a relatively late manifestation of ID, and recent studies have highlighted the high prevalence of ID without anaemia in patients with IBD [10,11,12]. ID without anaemia in patients with IBD is underdiagnosed and untreated, despite causing an array of clinical symptoms, such as fatigue, sleeping disorders, attention deficit and agitation, that have negative effects on HRQoL. However, the decision to supplement with oral iron in patients with IBD without anaemia is not straightforward, as non-absorbed ferrous oral iron has the potential to worsen IBD symptoms and aggravate intestinal inflammation. Thus, the clinical scenario, the patient’s history and individual preference should be taken into consideration [13,14].

While the criteria for diagnosing anaemia are well established (Hb < 13 g/dL in men and Hb < 12 g/dL in women) [15], diagnosing ID can be more challenging. Patients with ID have normal Hb levels and present with symptoms that can be associated with a multitude of conditions, ranging from hypothyroidism to depression [16]. In patients with IBD, the diagnostic criteria for ID depend on the level of inflammation. For patients with IBD in remission (without biochemical or clinical evidence of inflammation), serum ferritin < 30 µg/L and transferrin saturation index (TSAT) < 16% are indicative of ID [14].

It is clear from other disease areas including chronic heart failure [17] and chronic kidney disease [18] that treating ID can improve patients’ HRQoL. Two small studies have shown that after 6 and 12 weeks, intravenous administration of iron sucrose or iron polymaltose improves HRQoL in nonanaemic but iron-deficient patients with IBD [19,20]. To the best of our knowledge, no other studies have investigated whether the correction of ID with FCM can rapidly impact the HRQoL of patients with IBD.

The aim of this study was to determine whether treating ID in patients with IBD without anaemia with a single dose of FCM was an effective, safe and fast approach for reducing the symptoms of ID and improving patient HRQoL.

## 2. Materials and Methods

### 2.1. Study Design and Patient Characteristics

This was a multicentre, prospective, observational study to assess whether a single 500 mg dose of FCM (Ferinject^®^, Vifor France, Paris, France) led to an improvement in the HRQoL of patients with IBD and ID without anaemia at 1 month after treatment. The safety and efficacy of FCM in restoring iron levels in these patients was also assessed. The study was carried out in accordance with the Declaration of Helsinki after approval of the protocol by the ethics committee at the General University Hospital of Valencia, Valencia, Spain.

Patients were recruited consecutively from the Digestive Disease Unit outpatient clinic in the General University Hospital of Valencia and six other participating centres. All patients gave their written informed consent.

The diagnosis of CD and UC was established by clinical, radiological, histological and endoscopic criteria [21], and disease severity was classified using the Montreal classification [22]. The partial Mayo score [23] and the Harvey–Bradshaw index [24] were used to define clinical activity in UC and CD, respectively, both before FCM infusion and 1 month after treatment. Analytical parameters (i.e., C-reactive protein (CRP) and calprotectin) and endoscopy results were considered if available in the pre- and post-treatment period [25].

To be eligible for the study, outpatients had to be in remission or have mild-to-moderate clinical activity (Harvey–Bradshaw index ≤ 7 or partial Mayo score ≤ 4), ID defined by a low ferritin analytical profile (<30 μg/L in the absence of inflammatory activity or <100 μg/L with inflammation) and/or a low TSAT (<16%). In addition, Hb levels had to be ≥13 g/dL in adult men and ≥12 g/dL in adult women to confirm the absence of anaemia. All included patients had shown intolerance to or lack of efficacy of prior treatment with oral iron in any of its formulations (i.e., ferrous or ferric salts).

The following patients were excluded: patients with disease that could contribute to ID (severe cardiopulmonary, hepatic or renal); patients with active moderate-to-severe IBD activity (Harvey–Bradshaw index ≥ 8 or partial Mayo score ≥ 5); patients taking nonsteroidal anti-inflammatory drugs during the 3 months prior to inclusion in the study or taking anticoagulant drugs; patients taking FCM for diseases other than IBD; patients who had been given intravenous iron, erythropoietin or blood transfusion in the 3 months prior to the study; patients who were allergic to or intolerant of FCM or any of its excipients; women who were pregnant; patients who had started or modified treatment with antidepressants or anti-anxiety drugs one month before or during the study; patients with surgical resection in the previous year; and patients who were <18 years of age. No patient who received modifications to their treatment was able to continue in the study (either due to a flare-up of activity or by decision of the clinician) to avoid possible bias.

Clinical and demographic data were collected from patient records. Analytical profiles (i.e., Hb, serum iron (s-iron), serum ferritin (s-ferritin), TSAT, serum vitamin B12 and folic acid) and completed quality-of-life questionnaires (Spanish versions of the Short Form Health Survey, version 2 (SF-12v2) and EuroQoL 5-Dimensions (EQ-5D)) [26,27] were collected before FCM infusion and 1 month after. Patients were also asked to report any ID-associated symptoms.

### 2.2. Administration of Intravenous Iron

Patients received a single 500 mg dose of FCM for the treatment of ID without anaemia on an outpatient basis. FCM was administered as indicated in the summary of product characteristics (SmPC) [28]. The patients remained under hospital observation for ≥30 min after the end of the FCM administration.

### 2.3. Aims of the Study

The primary objective was to evaluate early changes in HRQoL in patients with IBD with ID without anaemia after the administration of FCM.

Secondary objectives were to evaluate the biological response to FCM administration using iron parameters, identify changes in symptoms reported by patients after administration of FCM and detect and describe the possible adverse events (AEs) of treatment with FCM.

### 2.4. HRQoL Assessment

Changes in HRQoL indices were evaluated to determine whether the patients’ perceived health had improved 1 month after receiving FCM. Patients were asked to complete two questionnaires, the SF-12v2 and the EQ-5D, both before and 1 month after the FCM infusion.

The SF-12v2 is one of the most widely used instruments for assessing self-reported HRQoL in patients with chronic disease. It consists of 12 questions that measure 8 physical and mental health domains: general health, physical functioning, role physical, body pain, vitality, social functioning, the role of emotions and mental health. Scores range from 0 to 100, with 0 indicating the lowest level of health that can be measured and 100 indicating the highest level of health that can be measured. The EQ-5D is a preference-based measure of health status that is routinely applied in clinical trials, population studies and real-world clinical settings. It consists of five questions to assess mobility, self-care, usual activities, pain/discomfort and anxiety/depression as well as an evaluation portion in which respondents evaluate their overall health status on a visual analogue scale ranging from 0 (which corresponds to “the worst health you can imagine”) to 100 (“the best health you can imagine”).

Patients were asked about the presence of self-reported ID-associated symptoms such as fatigue, dyspnoea, paleness, weakness/asthenia, headaches, dizziness, tachycardia and hair loss both before and 1 month after FCM infusion.

### 2.5. Efficacy and Safety Analysis

Response to FCM was defined as the normalization of iron parameters (s-ferritin > 100 µg/L and TSAT > 15%) at 1 month after FCM infusion. To identify any AEs associated with FCM, a clinical evaluation and physical examination were carried out, and vital signs were recorded either during or shortly after the infusion. One month later, patients were asked whether they had experienced any AEs in the 24 h after the infusion.

### 2.6. Statistical Analysis

The results of continuous/quantitative variables are presented using the arithmetic mean and standard deviation (SD). For categorical/qualitative variables, the results are shown using frequencies and percentages. Numerical variables with a nonnormal distribution are presented using the median and interquartile range.

Comparisons between pre- and post-FCM administration were performed using the Wilcoxon signed rank test for nonnormally distributed variables and McNemar’s test and an exact binomial test for paired data. For correlations, the Pearson correlation coefficient was calculated. The calculations were performed with a significance level (α) of 0.05, a statistical power (1−β) of 0.80, an effect size (f) of 0.4, an odds ratio of 2.5 and a proportion of discordant pairs of 0.3. The results indicated the need for a sample size (n) of 93 subjects. The statistical analysis was performed using the software package R 4.0.3 for Windows [29].

## 3. Results

### 3.1. Patient Population

Of the 105 patients who received a single 500 mg infusion of FCM, 98 patients (70 females and 28 males) completed the study (Figure 1). Seven patients were excluded from the analyses because no HRQoL data were available, one of them presented a mild hypersensitivity reaction and another patient presented a clinical relapse which was treated with corticosteroids. The mean ± SD age of patients was 43 ± 12.41 years. The IBD diagnosis was CD in 66 patients and UC in 32 patients. The mean (SD) time from IBD diagnosis to study enrolment was 10.86 (6.91) years. The baseline patient characteristics are summarised in Table 1. Most patients (87.76%) were receiving at least one drug for IBD treatment.

### 3.2. Efficacy and Safety of FCM on ID

After 1 month, a single dose of FCM increased iron parameters to normal levels; s-ferritin increased from 48.4 to 175 μg/L, s-iron from 51.9 to 84.4 μg/L, and TSAT from 12.8% to 27.2% (Wilcoxon test, *p* < 0.001). The percentage of patients with TSAT < 16% was also significantly reduced from 73.75% (59 patients) before treatment to 8.75% (7 patients) after treatment with FCM (McNemar test, *p* < 0.001). The effects of FCM on iron parameters are shown in Table 2. FCM had similar effects on iron parameters in patients with CD (n = 66) and UC (n = 32). No changes from the baseline in the patients’ clinical activity were observed at 1 month after FCM treatment.

Eight of the one hundred and five patients (7.6%) who received FCM had at least one AE. One patient (0.9%) had a mild hypersensitivity reaction that required discontinuation of treatment. This patient experienced generalised erythema, nasal congestion, nausea and mild hypotension. The other seven patients experienced mild AEs that did not require discontinuation of the drug, including irritation at the infusion site (*n* = 1), mild rash (*n* = 2), nonspecific dizziness (*n* = 5) and nausea and vomiting (*n* = 1).

### 3.3. Inflammatory Biomarkers and Other Biochemical Parameters

The mean (SD) of CRP and faecal calprotectin (as markers of inflammation) remained unchanged before and one month after iron infusion: CRP 4.12 (6.16) mg/L vs. 3.09 (5.10) mg/L (*p* > 0.05) [normal CRP value < 5 mg/L]; faecal calprotectin 333.21 (647.64) µg/g vs. 290.10 (510.22) µg/g (*p* > 0.05) [normal faecal calprotectin value < 200 µg/g].

Regarding the means (SD) of vitamin B12 and folic acid, they remained stable before and one month after the iron infusion: vitamin B12 346.91 (270.79) pg/mL vs. 340.10 (272.25) pg/mL (*p* > 0.05) [normal vitamin B12 range 180–914 pg/mL]; folic acid 7.01 (3.34) ng/mL vs. 6.09 (3.30) ng/mL (*p* > 0.05) [normal folic acid range 3–17 ng/mL].

### 3.4. Impact of FCM on Self-Reported ID Symptoms

A statistically significant improvement was found in all the self-reported ID symptoms except for paleness (Figure 2a). The greatest improvements were in weakness/asthenia, fatigue, dizziness and hair loss. No post-treatment ID symptoms were reported by 72.45% of the patients, which represents an improvement in almost 50% of patients. The percentage of patients reporting symptom improvement was similar between men (46.43%) and women (47.14%).

After FCM treatment, 72.73% of patients with CD and 71.88% of patients with UC reported not having any ID symptoms (Figure 2b,c).

### 3.5. Impact of FCM on Patients’ HRQoL (EQ-5D and SF-12v2)

EQ-5D questionnaire analysis shows that significantly fewer patients reported experiencing problems carrying out their usual activities, pain/discomfort or anxiety/depression (*p* < 0.005) after treatment with FCM. Over 39% reported having no problems after treatment (Table 3).

Over 41.51% of patients with CD and 35.48% of patients with UC reported having no problems after treatment.

Patients reported a better state of health after FCM administration, as shown by improvements in the mean EQ-5D scores for all patients (Table 4) and in the improvement of the mean scores on the EQ-5D visual analogue scale (Table 5). Similar improvements were observed in the mean (SD) EQ-5D scores in patients with CD (*p* < 0.01) and in those with UC (*p* < 0.05). The EQ-5D visual analogue scale scores in patients with CD showed an increase after treatment, but it was not statistically significant (*p* = 0.057). The difference was statistically significant in patients with UC (*p* < 0.01).

The SF-12v2 questionnaire scores also indicated that patients reported overall better physical and mental health status after FCM treatment. The mean (SD) physical score increased from 42.2 (10.66) before treatment to 45.1 (10.68) after treatment (*p* < 0.01). The mean (SD) mental score increased from 42.2 (13.08) pretreatment to 49.5 (12.23) posttreatment (*p* < 0.001) (Table 6). In patients with CD, there were statistically significant differences between the pre- and post-FCM SF-12v2 mental and physical scores (*p* < 0.05). In patients with UC, there were statistically significant differences between the pre- and post-FCM SF-12v2 mental scores (*p* < 0.001), and although there was an increase in the physical scores, it did not reach statistical significance (*p* = 0.06) (Table 6).

## 4. Discussion

In 98 patients with IBD and ID without anaemia, a single 500 mg dose of FCM restored iron parameters and improved patients’ symptoms and HRQoL. A clear improvement was observed in the analytical values of s-ferritin and TSAT at 1 month after FCM administration. This clinical benefit translated into a statistically significant improvement in patients’ HRQoL regardless of sex and IBD condition (UC or CD).

FCM is a safe and effective treatment for ID and iron deficiency anaemia, chronic heart failure and chronic kidney disease [30,31,32,33,34]. FCM has also been shown to be safe and effective for the treatment of anaemia in patients with IBD [35] to prevent the recurrence of anaemia in IBD [36] and to improve the HRQoL of these patients [8,37]. FCM treatment is also recommended for the management of ID and anaemia by the European Crohn’s and Colitis Organisation consensus statement [14]. However, the effects of FCM on HRQoL in iron-deficient patients with IBD without anaemia have not been reported until now.

ID in the absence of anaemia negatively impacts normal perception of HRQoL in patients with IBD who are in remission. The correction of ID could be a new target in the treatment of these patients [11]. The early diagnosis and treatment of ID without anaemia to improve patient HRQoL, as suggested by Peyrin-Biroulet and colleagues [38], represents a paradigm shift in the management of ID without anaemia [39].

Only two studies have examined the effects of treating ID in patients with IBD without anaemia using different formulations of intravenous iron and with much later post-administration evaluation times. In 2017, a study of 13 patients with hypoferritinaemia without anaemia found that a single dose of intravenous iron polymaltose significantly improved HRQoL after 6 weeks [20]. Another study in 2015 showed improvements in the HRQoL of 85 nonanaemic but iron-deficient patients with IBD 12 weeks after administration of intravenous iron sucrose [19].

The increases in s-ferritin, s-iron and TSAT that we observed in this study are comparable with those in other studies in which FCM was administered to improve iron parameters across a range of conditions [40]. Importantly, our findings indicate that after 1 month, the recommended time for assessing the response to FCM (13,28), a single dose of FCM significantly improves patients’ symptoms, with nearly 50% of patients reporting no ID symptoms as well as improved HRQoL. We found consistent improvements in the EQ-5D and SF-12v2 questionnaire scores, as well as in the overall health status of the patients.

We did not use the IBDQ or other disease-specific questionnaires to assess the patient HRQoL since treating ID may not improve IBD activity-related symptoms such as diarrhoea, abdominal pain and rectal bleeding and thus may not reflect an improvement in a patient’s HRQoL.

Our patients’ responses to treatment were not influenced by disease activity since they were all in remission or had mild IBD activity. The evaluation to determine presence of inflammation was based on faecal calprotectin, PCR and endoscopy. Due to the low number of patients for whom endoscopy was available (based on routine clinical practice), it was not taken into account in the analysis. The CRP values remained within normal range both before and after iron infusion. Faecal calprotectin was observed to be elevated in a small number of patients, with no elevation occurring in the month after treatment. Due to the short follow-up period (1 month), no treatment changes were made on the decision of the clinician based on the isolated determination of faecal calprotectin. This is due to the fact that in clinical practice, the determination of calprotectin was repeated one month after the baseline determination and at that time, a decision to makes changes to the treatment either could or could not be made.

There were a few limitations to this study. First was the relatively small number of patients with CD and UC, which meant that although the effects of FCM on HRQoL were statistically significant when both groups were assessed together, when they were analysed separately, no statistically significant improvement was observed in the EQ-5D visual analogue scale ratings in patients with CD or in the SF-12v2 physical scores in patients with UC after treatment. Another limitation was the lack of a control group, such as a group of patients receiving oral iron supplements; however, as many patients do not respond to or experience gastrointestinal AEs with oral iron, it would be difficult to conduct this type of study with such a control group. In this situation, the alternative would have been to use a placebo in the control group. It is also worth noting that the patients’ responses to FCM could have been influenced by medication related to IBD or treatment for other conditions (such as depression/anxiety).

Future studies should examine the long-term effects of FCM for the treatment of ID in patients with IBD to determine whether higher doses are required to maintain both the clinical benefits and HRQoL improvements that we observed after 1 month. Overall, the results of this study support the inclusion of FCM into treatment schemes and guidelines for iron-deficient patients with IBD without anaemia.

## 5. Conclusions

In conclusion, this study demonstrates the clinical benefits of a single dose of FCM in rapidly restoring the iron parameters, as well as ID symptoms and HRQoL, in nonanaemic patients with IBD with ID. Given the increasing prevalence of this complex disease, early diagnosis of ID and administration of FCM could contribute to reducing the social and economic burden of IBD.

## Figures and Tables

**Figure 1 jcm-11-02786-f001:**
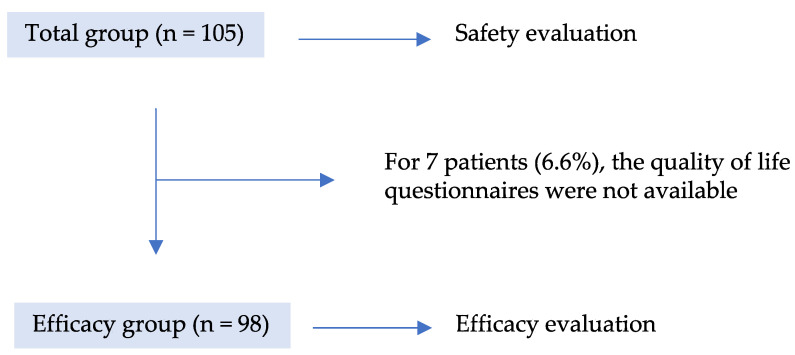
Patient disposition.

**Figure 2 jcm-11-02786-f002:**
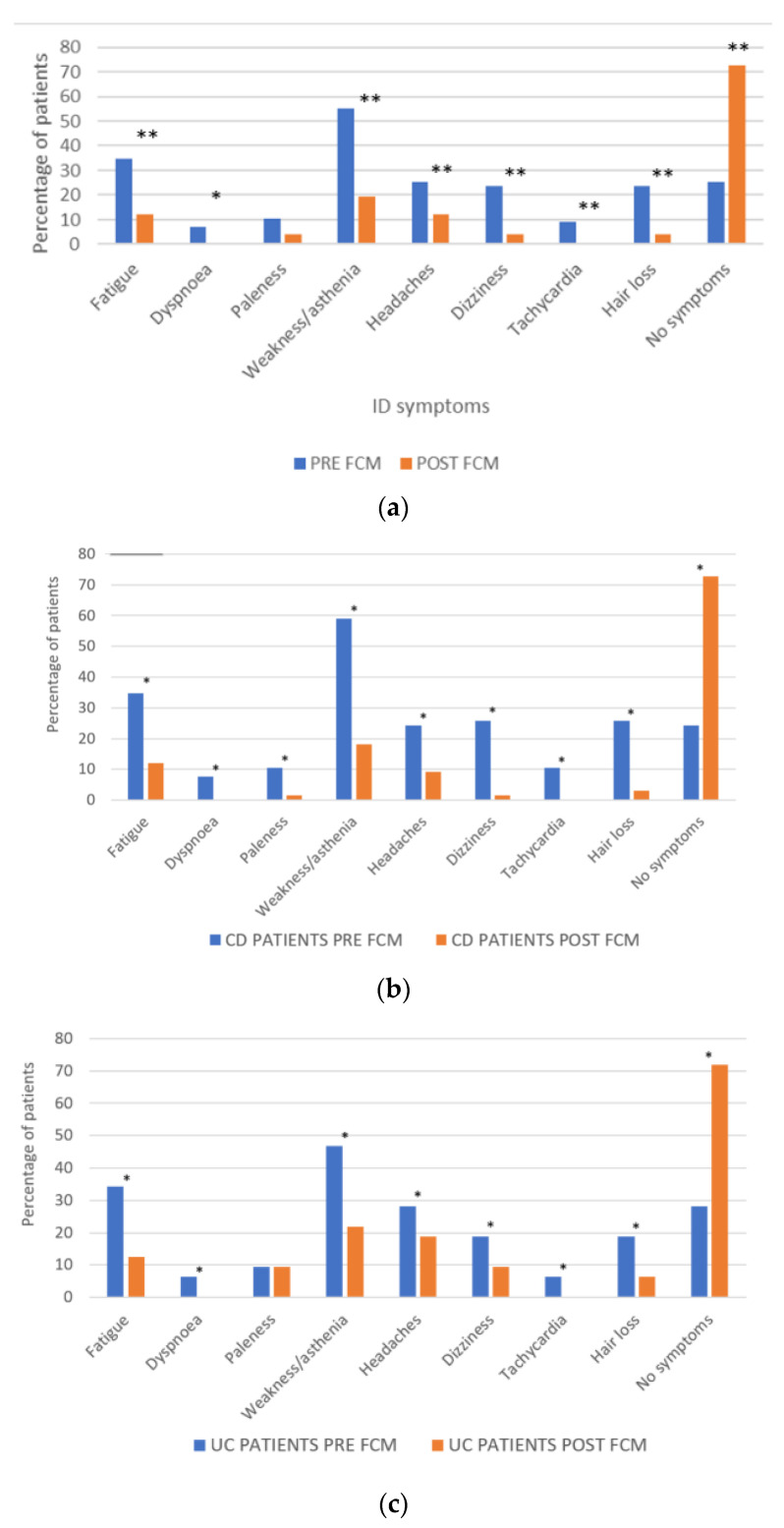
(**a**) Effect of FCM on self-reported ID symptoms; * *p* < 0.05; ** *p* < 0.01 (McNemar test). (**b**) Effect of FCM on ID symptoms patients with CD; * *p* < 0.05 (McNemar test) (**c**) and UC; * *p* < 0.05 (McNemar test).

**Table 1 jcm-11-02786-t001:** Patient baseline characteristics.

Characteristic	n = 98
Mean age (SD), years	43 (12.41)
Gender Female, n (%)	70 (71.43)
Type of disease, n (%)	
Crohn’s disease	66 (67.35)
Ulcerative colitis	32 (32.65)
Harvey–Bradshaw index, n (%)	
Remission (score 0–4)	53 (54.07)
Mild (score 5–7)	13 (13.26)
Moderate (score 8–16)	0
Severe (score > 16)	0
Partial Mayo Scoring Index; n (%)	
Remission (score 0–1)	18 (18.36)
Mild (score 2–4)	14 (14.32)
Moderate (score 5–6)	0
Severe (score 7–9)	0
Previous surgery, n (%)	
Yes	26 (26.53)
No	72 (73.47)
Mean CRP (SD), mg/L ^a^	4.12 (6.16)
Mean faecal calprotectin (SD), µg/g ^b^	333.21 (647.64)
Mean Hb (SD), g/dL	13.44 (1.02)
Mean s-ferritin (SD), μg/L	48.38 (64.60)
Mean s-iron (SD), μg/dL	51.92 (41.03)
Mean TSAT (SD), %	12.78 (4.19)
Mean vitamin B12 (SD), pg/mL ^c^	346.91 (270.79)
Mean folic acid (SD), ng/mL ^d^	7.01 (3.34)
Treatment, n (%)	
Mesalazine	31 (31.63)
Steroids	6 (6.12)
Thiopurines	24 (24.49)
Methotrexate	4 (4.08)
Anti-TNF	38 (38.78)
Vedolizumab	6 (6.12)
Ustekinumab	11 (11.22)
Apheresis	2 (2.04)

CRP = C-reactive protein; SD = standard deviation; s-ferritin = serum ferritin; s-iron = serum iron; TSAT = transferrin saturation index; TNF = tumour necrosis factor; ^a^ = Normal CRP value < 5 mg/L; ^b^ = Normal faecal calprotectin value < 200 µg/g; ^c^ = Normal vitamin B12 range 180–914 pg/mL; ^d^ = Normal folic acid range 3–17 ng/mL.

**Table 2 jcm-11-02786-t002:** Effects of ferric carboxymaltose administration on iron parameters.

**All Patients (n = 98)**	**Pre-FCM**	**Post-FCM**	**95% CI**	** *p* ** **-Value ***
Mean s-ferritin, μg/L n (pre/post) ^1^ = 87	48.4	175.0	126.6 (97.5, 135)	<0.001
Mean s-iron, μg/L n (pre/post) ^1^ = 84	51.9	84.4	32.5 (28.5, 44)	<0.001
Mean TSAT, % n (pre/post) ^1^ = 75	12.8	27.2	14.4 (11.6, 17)	<0.001
**Patients with CD (n = 66)**	**Pre-FCM**	**Post-FCM**	**95% CI**	** *p* ** **-Value ***
Mean s-ferritin, μg/L n (pre/post) ^1^ = 56	44.7	175.0	130.3 (94.5, 142.5)	<0.01
Mean s-iron, μg/L n (pre/post) ^1^ = 51	47.6	76.3	28.7 (22.0, 38.5)	<0.01
Mean TSAT, % n (pre/post) ^1^ = 44	13.1	24.9	11.8 (11.6, 17.0)	<0.01
**Patients with UC (n = 32)**	**Pre-FCM**	**Post-FCM**	**95% CI**	** *p* ** **-Value ***
Mean s-ferritin, μg/L n (pre/post) ^1^ = 28	56.4	174.8	118.4 (78.5, 154.0)	<0.01
Mean s-iron, μg/L n (pre/post) ^1^ = 28	60.6	99.9	39.3 (33.5, 65.5)	<0.01
Mean TSAT, % n (pre/post) ^1^ = 26	12.1	31.2	19.1 (11.6, 17.0)	<0.01

CI = confidence interval; CD = Crohn’s disease; FCM = ferric carboxymaltose; s-ferritin = serum ferritin; s-iron = serum iron; TSAT = transferrin saturation; UC = ulcerative colitis; ^1^ = Number of patients with laboratory values before and after FCM treatment; * = Wilcoxon signed-rank test.

**Table 3 jcm-11-02786-t003:** EQ-5D descriptive system.

Problem (% of Patients)	Pre-FCM	Post-FCM	Difference	*p*-Value *
Mobility Self-care	23.40	16.67	–6.74	0.194
	9.57	3.57	–6.00	0.136
Usual activities	55.32	35.71	–19.60	0.002
Pain/discomfort	70.21	53.57	–16.64	0.003
Anxiety/depression	55.32	28.57	–26.75	<0.001
With no problems	17.02	39.29	+22.26	0.008

EQ-5D = EuroQoL 5-Dimensions; FCM = ferric carboxymaltose; * = McNemar test and an adaptation of the exact binomial test.

**Table 4 jcm-11-02786-t004:** Mean EQ-5D scores (SD).

	Pre-FCM	Post-FCM	*p*-Value *
All	0.721 (0.283)	0.823 (0.236)	<0.001
Women	0.692 (0.294)	0.807 (0.240)	<0.001
Men	0.792 (0.243)	0.865 (0.225)	na
CD	0.694 (0.320)	0.816 (0.249)	<0.01
UC	0.773 (0.187)	0.835 (0.216)	0.02

EQ-5D = EuroQoL 5-Dimensions; SD = standard deviation; FCM = ferric carboxymaltose; * = Wilcoxon test; na = not applicable due to sample size; CD = Crohn’s disease; UC = ulcerative colitis.

**Table 5 jcm-11-02786-t005:** Mean EQ-5D visual analogue scores (SD).

	Pre-FCM	Post-FCM	*p*-Value *
All	62.6 (18.8)	70.1 (18.8)	<0.01
Women	61.6 (18.9)	69.0 (19.3)	<0.01
Men	64.8 (18.7)	73.0 (17.6)	na
CD	62.1 (19.6)	69.4 (20.2)	0.057
UC	63.4 (17.3)	71.4 (16.9)	<0.01

EQ-5D = EuroQoL 5-Dimensions; SD = standard deviation; FCM = ferric carboxymaltose; * = Wilcoxon test; na = not applicable due to sample size; SD, standard deviation; CD = Crohn’s disease; UC = ulcerative colitis.

**Table 6 jcm-11-02786-t006:** SF-12v2 scores before and after administration of FCM.

	Mean SF-12v2 Scores (SD)		
All Patients	Pre-FCM	Post-FCM	*p*-Value *
Physical score	42.2 (10.66)	45.1 (10.68)	0.002
Mental score	42.2 (13.08)	49.5 (12.23)	<0.001
**CD**			
Physical score	41.7 (11.0)	44.3 (10.5)	0.01
Mental score	42.7 (12.2)	48.4 (13.1)	<0.001
**UC**			
Physical score	43.2 (10.1)	46.6 (11.1)	0.06
Mental score	41.1 (14.9)	51.5 (10.5)	<0.001

SF-12v2 = 12-item Short-Form Health Survey, version 2; FCM = ferric carboxymaltose; SD = standard deviation; * = Wilcoxon test; CD = Crohn’s disease; UC = ulcerative colitis.

## Data Availability

All the data generated or analysed during this study are included in this article. Further enquiries can be directed to the corresponding author.

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
