# Peer review of "Ferric Carboxymaltose Improves the Quality of Life of Patients with Inflammatory Bowel Disease and Iron Deficiency without Anaemia"

_jcm, 2022, doi:10.3390/jcm11102786_

Round 1

Reviewer 1 Report

Thank you for inviting me to review, I hope I have contributed and I am available for any clarifications.

The article in question evaluated the effects of iron supplementation (ferric carboxymaltose) on quality of life and iron-related parameters in patients with inflammatory bowel disease with iron deficiency but without anaemia. It presents good writing and, using appropriate methodology, has brought important results, being ready for publication. Here are some considerations which I hope will contribute to the article.

Title: Adequate, related to the results obtained in the study.

Abstract: Adequate.

Introduction: Adequate but deserves a suggestion: The questionnaires used were cited but deserve a more detailed description, either in the introduction or in the methodology.

Material and method:
- It would be interesting to include how the presence of inflammation was assessed. The results describe the C-reactive protein, was it through it?
- Item 2.4 - Are the SF-12v2 and EQ-5D questionnaires specific to quality of life? Are they validated? What are the differences between what they evaluate? It would be interesting to think about briefly describing these questionnaires.

Results:
- Does table 1 show that some patients underwent previous surgery? How long after surgery they could be included in the study? It would be nice to describe this in the methodology;
- CRP, calprotectin, vitamin B12 and folic acid were measured in the pre-treatment period. It would be interesting to describe why they were evaluated only in the pre-treatment period, since these parameters were not discussed;
- Table 3: In order to make the table more self-explanatory, indicate in the legend which statistical test was used.

Discussion: Adequate and well related to most results. However, some of the parameters evaluated were not discussed. They are: CRP, calprotectin, vitamin B12 and folic acid.

Conclusion: Adequate and in accordance with the results.

Reviewer 2 Report

This well written and interesting paper by Huguet et al describes the clinical benefit and safety of a single dose of IV iron administration in IBD patients with iron deficiency and normal hemoglobin levels. This is of interest to the clinical gastroenterologist and indeed a question that arises often in clinical practice. 

I do have some comments which I feel should be addressed to a greater degree in the manuscript: 

  • The authors repeatedly comment that patients were in remission or had mild disease activity. The caveat was that this was defined by clinical scores alone and not objective markers of inflammation or endoscopy. Over 30% of patients without symptoms have active disease and this may affect there general well being. There is no mention as to how this was accounted for. This is especially important considering the fecal calprotectin was significantly increased in the cohort at baseline (a mean of 331). This should at the very least be discussed.
  • There is no mention of whether IBD medications were changed/added or kept the same throughout the study period. This is important as this is a very significant potential confounder. Considering my above point is it really likely that no therapy changes were made in patients with high fecal calprotectins? This should be clarified in the text. 
  • Without a placebo control (beyond oral iron , as mentioned in the discussion) it is difficult to ascertain the true clinical benefit of a single iron dose beyond improving lab parameters.  
